# A Review of the Community Health Club Literature Describing Water, Sanitation, and Hygiene Outcomes

**DOI:** 10.3390/ijerph18041880

**Published:** 2021-02-15

**Authors:** Jason Rosenfeld, Ruth Berggren, Leah Frerichs

**Affiliations:** 1Center for Medical Humanities & Ethics, Joe R. and Teresa Lozano Long School of Medicine, University of Texas Health Science Center at San Antonio, 7703 Floyd Curl Dr, San Antonio, TX 78229-3900, USA; BerggrenR@uthscsa.edu; 2Department of Health Policy and Management, Gillings School of Global Public Health, University of North Carolina at Chapel Hill, 1102C McGavran-Greenberg, Chapel Hill, NC 27599-7411, USA; leahf@email.unc.edu

**Keywords:** water, sanitation and hygiene, community health clubs, health promotion, behavior change, community-based

## Abstract

The Community Health Club (CHC) model is a community-based health promotion program that utilizes water, sanitation, and hygiene (WASH) education as the first stage of a longitudinal development process. Although the CHC model has been implemented in fourteen countries over 20 years, this is the first review of the literature describing the model’s outcomes and impact. We conducted a review of the literature that provided quantitative or qualitative evidence of CHC interventions focused on WASH in low- and middle-income countries. We identified 25 articles that met our inclusion criteria. We found six major outcomes: WASH behaviors and knowledge, social capital, collective action, health, and cost or cost-effectiveness. The most consistent evidence was associated with WASH behaviors and knowledge, with significant effects on defecation practices, hand washing behaviors, and WASH knowledge. We also found qualitative evidence of impact on social capital and collective action. CHCs catalyze favorable changes in WASH behaviors and knowledge, yielding outcomes commensurate with other WASH promotion strategies. This review provides insights into the model’s theory of change, helping identify areas for further investigation. The CHC model’s holistic focus and emphasis on individual and collective change offer promising potential to address multiple health and development determinants.

## 1. Introduction

The impact of inadequate water, sanitation, and hygiene (WASH) is significant. Approximately 4.2% of global morbidity is attributable to inadequate WASH, the majority of which is associated with infectious diarrhea, which kills more children under the age of five each year than HIV/AIDS, tuberculosis, and malaria combined [1,2,3]. Although global access to clean water and safe sanitation has improved since 1990, 785 million people remain without access to basic drinking water sources, 2 billion people remain without access to basic sanitation services, and 3 billion people still lack basic hand washing facilities at home [4]. Furthermore, progress has been uneven, with least developed countries lagging and a growing disparity between urban and rural communities [4]. One approach to improve WASH, the Community Health Club (CHC) model, is a participatory, community-based program that holds promise because of its wide implementation and evaluation indicating positive outcomes.

Community health clubs (CHCs) are voluntary, community-based organizations that provide a forum for the dissemination of preventive health information and opportunities for consensus building, behavior change, and collective action [5]. The CHC model utilizes WASH education as the first of a four-stage integrated development process. Subsequent stages of the model vary by program, but may include nutrition, reproductive and sexual health, women’s empowerment, and income-generating activities. Under the first stage, the CHC model is designed to alter WASH practices and create demand for water and sanitation infrastructure by leveraging the power of the peer group and promoting community cohesion [6]. CHCs provide a forum for experiential learning, discussion, and problem solving that draws upon multiple behavioral theories, including the health belief model, the theory of reasoned action and planned behavior, and social learning theory, and is set within the conceptual frameworks of participatory development and social capital [6,7].

CHC theory and practice have been described in more detail elsewhere [5,6,7,8,9]. In brief, the CHC model aims to bridge the gap in WASH knowledge and behaviors by creating a peer group dedicated to learning and taking collective action to improve the health of their community [5,6]. CHCs aim to facilitate changes in communal norms and values by promoting community cohesion, creating collective knowledge, and building group consensus. The model recognizes that knowledge is necessary for WASH behavior change but is insufficient without a critical mass providing individuals an incentive to change based upon the need for group conformity. To normalize new behaviors and achieve shared values, CHCs meet weekly at a regular venue where people explore their individual and collective knowledge using participatory health education techniques. These techniques encourage people to engage with new ideas and compare traditional knowledge and beliefs with current scientific knowledge and safe practices. This process of self-discovery, facilitated by a trained community health worker, rather than top-down transfer of knowledge through expert advice, is essential to the creation of new knowledge and achievement of sustainable behavioral changes and collective action.

Although several WASH education and behavior change interventions have been implemented and researched over the past 25 years (e.g., participatory hygiene and sanitation transformation, community-led total sanitation, sanitation marketing, total sanitation), the CHC model merits special attention due to its holistic WASH focus (e.g., not just hand washing or latrine construction) and relatively widespread dissemination [10]. Since 1994, over 3000 CHCs have been formed in low- and middle-income countries, benefiting over two million people across Africa, Asia, and the Caribbean. To date, CHC programs focused primarily on WASH have been implemented in both rural and urban communities in Zimbabwe, Uganda, South Africa, Rwanda, Sierra Leone, Tanzania, Kenya, Burkina Faso, the Democratic Republic of Congo, Namibia, Haiti, Dominican Republic, Vietnam, and Papua New Guinea [11].

Despite widespread use of the CHC model across different contexts over the past 20 years, gaps remain to strengthen and improve its implementation and evaluation. Apart from Zimbabwe, where CHCs originated, the model has only been implemented and evaluated at scale in Rwanda [12]. To our knowledge, the peer-reviewed literature has never been systematically reviewed in order to synthesize and describe the impact of the model. Therefore, we conducted a comprehensive literature review of CHC programs, focused specifically on the first stage of WASH promotion. Our specific aims were to: (1) identify the commonly reported outcomes of CHC WASH programs and (2) synthesize the impact of CHC programs by each outcome. This review expands our understanding of the outcomes of a commonly used WASH promotion intervention, thereby allowing practitioners and researchers alike to more effectively compare outcomes across WASH promotion interventions. This manuscript also synthesizes the WASH outcomes most commonly associated with the CHC model, informing our understanding of this model’s theory of change and enhancing future evaluations of this intervention.

## 2. Materials and Methods

We conducted a review of the CHC literature using the PubMed, Global Health, Scopus, and Google Scholar electronic databases. We identified additional studies and manuscripts through expert recommendations and by mining the citations of relevant articles discovered through the database search. The search terms for our database search strategy are provided in Table 1. The final search was completed on 28 April 2020.

To be included in this review, studies and papers had to: (1) describe and evaluate CHC interventions focused only on WASH in low- and middle-income countries; (2) provide empirical and/or qualitative evidence of the impact of a CHC intervention; and (3) be published in English. Due to limited peer-reviewed evidence describing the impact of CHC WASH interventions, this review included published papers, program reports, conference proceedings, and working papers. Studies and papers were excluded if: (1) the focus of the CHC intervention was not WASH; (2) the intervention was implemented in a high-income country; and (3) a study referenced or utilized previously reported monitoring or evaluation data. For studies that referenced previously reported data, we retained the document that reported the data first and excluded all subsequent citations.

We first screened articles by title, abstract, and keywords to remove clearly irrelevant articles. We then reviewed the full text of the remaining articles for adherence to our inclusion and exclusion criteria. The following information was then abstracted from each of the included articles: type of article (e.g., peer-reviewed journal article, doctoral dissertation, master’s thesis, conference paper, working paper, and programmatic report), study design or methods, reported outcomes, and country. Data were extracted into Microsoft OneNote for Windows 10 (Version 16001.13328.20478.0) and Microsoft Excel for Microsoft 365 (Version 13127.21064), Redmond, WA, USA.

## 3. Results

Twenty-five citations were included in this review. Figure 1 shows the PRISMA diagram of the study selection process. The characteristics of the articles, including the main outcomes measured, are presented in Table 2, while Table 3 presents the article’s study samples, measures, and results. Our review included five peer-reviewed research articles [6,7,8,12,13], seven conference papers [14,15,16,17,18,19,20], six master’s theses [21,22,23,24,25,26], four program reports [27,28,29,30], two doctoral dissertations [5,9], and one working paper [31]. Of the 25 articles reviewed, eleven used a cross-sectional study design [5,6,7,14,16,17,22,23,24,25,27], five utilized a quasi-experimental design [9,18,19,20,31], three utilized a case study design [8,13,21], three reported qualitative data abstracted from field reports and site visits [28,29,30], two utilized a time series design [15,26], and one used a cluster randomized controlled study design [12]. Of the 24 non-randomized designs, 14 included a comparison group [5,6,7,8,9,12,13,19,20,21,22,24,27,31] and 14 utilized mixed methods [5,6,7,8,9,13,19,20,21,22,23,25,27,31]. A majority of articles (*n* = 21) described CHC interventions in Africa, 11 of which were from Zimbabwe [5,6,7,13,15,17,18,21,22,23,25], followed by four from Rwanda [12,20,24,26], three from South Africa [16,18,19], three from the Democratic Republic of Congo [28,29,30], two from Uganda [14,17], and one from Sierra Leone [27]. The remaining described programs from the Caribbean island of Hispaniola in Haiti [8,9] and the Dominican Republic [31]. Finally, the articles in this review reported outcomes of CHC interventions in six categories: most (*n* = 23) reported WASH behavioral outcomes [5,6,7,8,9,12,13,14,16,17,18,19,20,21,22,23,24,25,26,27,28,29,31], about half (*n* = 13) reported changes in WASH knowledge [5,7,8,9,21,22,23,25,27,28,29,30,31], and ten reported outcomes associated with both social capital [5,6,7,8,9,13,21,22,27,29] and collective action [8,9,16,19,20,21,22,23,27,28], while six reported health outcomes [12,15,27,28,29,30] and four reported cost and cost-effectiveness outcomes [5,17,18,24].

### 3.1. Behavior

All but two studies included in this review reported WASH behavioral outcomes. Nineteen studies reported quantitative results [5,6,7,8,9,12,13,14,16,17,18,19,20,21,22,24,25,26,31], while four provided qualitative or observational data [23,27,28,29] related to WASH behavioral changes. We categorized behavioral outcomes into the following dimensions: sanitation, hand washing, composite WASH behavioral scores, drinking water, and other WASH practices (including environmental management, kitchen hygiene, and personal hygiene).

#### 3.1.1. Sanitation

The majority of studies (*n* = 15) reported sanitation outcomes, which we further categorized into latrine construction and ownership [7,12,13,14,17,20,21,28,29], reduction in open defecation [7,8,9,13,19], and improved latrine hygiene [8,20,22,24,25] practices. Of these, 10 included a comparison group [7,8,9,12,13,19,20,21,22,24], six (40%) of which reported significant differences between samples in latrine construction and ownership [7,12,13,20], open defecation practices [7,8,13], and latrine hygiene [8,20]. One study reported a significant improvement in latrine hygiene behaviors from pre- to post-intervention within a sample of CHC participants [25]. Two studies from Rwanda, one randomized controlled trial (RCT) and one case–control study, reported that households who completed the full six-month WASH curriculum were significantly more likely to have a latrine post-intervention as compared to controls (RCT: 0.085, CI: 0.015–0.16, *p* = 0.017; CC: peri-urban: 89.4% vs. 74.2%, *p* = 0.0001; rural: 95.2% vs. 14.2%, *p* = 0.0001) [12,20]. In Zimbabwe, CHC participants were significantly more likely to practice safe sanitation (own a hygienic latrine and not practice open defecation) at final than comparison respondents (93.4% vs. 43.2%, *p* < 0.001) [7]. Finally, CHC participants in urban Haitian communities were significantly more likely than a comparison sample to report sharing a latrine with others rather than openly defecate (84.6% vs. 43.8%, *p* < 0.02) and have an observably clean latrine (74.4% vs. 38.9%, *p* < 0.001) [8].

#### 3.1.2. Hand Washing

Hand washing behaviors were the second most commonly reported outcome (*n* = 12), which we further categorized into observations of hand washing facilities [13,17,20,21,24,28,29] and hand washing practices [8,9,22,23,25]. Of these, seven included a comparison group [8,9,13,20,21,22,24] and two (17%) reported significant results [20,25]. A retrospective case–control study of the CHC program in Rwanda reported that participants who completed the full six-month WASH curriculum were significantly more likely than controls to have an observably functional hand washing facility (peri-urban: 74.2% vs. 13.7%, *p* = 0.0001; rural: 91.4% vs. 43.3%, *p* = 0.0001) and soap (peri-urban: 38.4% vs. 7.7%, *p* = 0.0001; rural: 92.4% vs. 4.2%, *p* = 0.0001) [20]. A study comparing a CHC and Community Led Total Sanitation (CLTS) program in Zimbabwe found that households in CHC communities were significantly more likely to own a hand washing facility (64% vs. 10%, *p* < 0.0001) six months after the intervention and were more likely to sustain use of that facility two years later (37% vs. 2%, *p* < 0.0001) [13]. Finally, one study from Zimbabwe explored a dose response of CHC participation and found a significant association between the number of CHC sessions attended and the use of soap during hand washing (χ^2^ = 30, df = 1, *p* < 0.0001) [25].

#### 3.1.3. Composite WASH Behavioral Scores

Eight of the studies reported grouped WASH behaviors [5,6,7,18,26] or composite scores of WASH behaviors [9,19,31]. These studies measured a range of 10–29 behaviors across five WASH dimensions: sanitation and defecation, drinking water, hand washing, kitchen hygiene, and environmental management. Six included a comparison group [5,6,7,9,19,31] and four (50%) reported significant results for grouped WASH behaviors [5,6,7,26]. None of the studies using composite scores reported significant results. In two studies of CHCs in Zimbabwe, the authors measured 20 observable indicators of good hygiene and reported that CHC households were significantly more likely than a comparison sample to practice 16 recommended WASH behaviors (*p* < 0.001) in one district, 9 recommended WASH behaviors (*p* < 0.01) in a second district, and 10 recommended WASH behaviors in a third district [6,7]. A retrospective analysis of program monitoring data from households in 50 communities that completed the full six-month WASH curriculum in Rwanda reported statistically significant increases in average hygiene scores (29 observable indicators) from baseline to six months (*p* = 0.01), after one year (*p* < 0.05), after two years (*p* < 0.05), and after three years (*p* < 0.05) and that 100% of hygiene indicators were observed in sample households three years after the intervention ended [26].

#### 3.1.4. Drinking Water

Seven studies reported changes in drinking water practices [8,12,16,21,23,24,25], specifically water treatment practices [8,12,16,24,25], use of safe or improved drinking water sources [16,21], and safe drinking water storage [8,23]. Four studies included a comparison group [8,12,21,24] and two (29%) reported significant results associated with drinking water treatment behaviors [12,25]. In the Rwandan RCT, the authors reported households that completed the full six-month WASH curriculum were significantly more likely than the control group to treat their drinking water (0.086, CI: 0.029–0.14; *p* = 0.003) [12]. In Zimbabwe, one study found a significant association between the number of CHC sessions attended and self-reported drinking water treatment behaviors (χ^2^ = 22.53, df = 1, *p* < 0.0001) [25].

#### 3.1.5. Other WASH Behaviors

Finally, nine studies reported WASH behaviors associated with environmental management [8,22,23,27,28,29], kitchen hygiene [14,17,21,27,29], and personal hygiene [14,17,21,23,27]. Of these studies, four included a comparison group [8,21,22,27] and none reported significant results. For environmental management, all studies reported observational data about the visible presence of garbage or of household garbage pits to manage solid waste. Two studies reported quantitative changes in environmental cleanliness, with one study from Zimbabwe reporting a 30–40% increase in observably clean yards in CHC households [22]. For kitchen hygiene, the studies reported observational data about the presence of pot racks for dish drying. In a study from Uganda, 58% of CHC participants were observed to have constructed a pot rack by project conclusion [17]. Finally, the studies reporting personal hygiene behaviors described quantitative and qualitative observations of bathing shelters and clotheslines or self-reported personal hygiene practices (e.g., teeth brushing, combing hair, bathing, cutting nails). In two studies from Uganda, the authors reported CHC participants constructed 6062 bathing shelters after four months, and after six months, 43% of CHC participants were observed to have constructed a bathing shelter [14,17].

### 3.2. Knowledge

The second most common outcome reported in the literature was WASH knowledge. Of the 13 studies that reported changes in WASH knowledge, six measured WASH knowledge quantitatively [5,7,8,9,22,31], while seven reported qualitative evidence about participants’ WASH knowledge [21,23,25,27,28,29,30]. All of the studies reporting quantitative results included a comparison sample, and five of the six reported significant increases in knowledge among CHC participants [5,7,8,9,31]. The studies that reported quantitative results measured respondents’ knowledge about the transmission or prevention of common WASH diseases (e.g., diarrhea, skin diseases, intestinal parasites, and malaria) and recommended WASH behaviors (e.g., hand washing, drinking water storage, composition of a homemade oral rehydration solution). Respondents were asked a series of four to nine questions and the number of correct responses was recorded. The total number of correct responses was then reported as a continuous variable [9,31], where higher scores equated to greater knowledge, or categorized as low, medium, or high knowledge [5,7,8,22]. Two studies analyzed the relationship between categorical WASH knowledge scores and WASH behaviors [5,7].

In the first Zimbabwe program, the author reported that CHC participants from two districts were significantly more likely to have “good” health knowledge (e.g., could describe the symptoms, transmission, and prevention of a disease) about nine questions about oral rehydration, diarrhea, malaria, bilharzia, worms, skin diseases, HIV/AIDS, TB, and child care than comparison respondents (*p* < 0.0001) [5]. A follow-up study describing the results from a third district found that 68.3% of CHC participants demonstrated “full knowledge” on the average of 10 topics compared to 38.2% of comparison respondents (*p* < 0.001) [7]. Similar results were reported in studies from Haiti. Urban CHC participants were significantly more likely to have “high” preventive WASH knowledge scores compared to a comparison sample from the same communities (71.2% vs. 4.1%, χ^2^ = 107.4, df = 3, *p* < 0.0001) [8]. In rural Haiti, the author analyzed the impact of the CHC intervention on composite WASH knowledge scores and found the intervention contributed to a 1.78-point increase in WASH knowledge scores from baseline to final in the CHC sample (CI (0.94, 2.62), *p* < 0.0001) [9]. The remainder of the studies reported qualitative changes in CHC participant WASH knowledge, one of which attributed CHC participants’ knowledge about the fecal oral route of diarrheal transmission to increases in latrine ownership and presence of hand washing facilities at the end of the program [21].

This relationship between WASH knowledge and behaviors was explored further by two studies from Zimbabwe. In two districts, 79% of CHC participants had “good” health knowledge and were significantly more likely to cover their drinking water container (*p* = 0.006), use a hand washing facility (*p* < 0.0001), have soap for hand washing (*p* < 0.0001), use the pour to waste method of hand washing (*p* < 0.0001), have a garbage pit (*p* = 0.01), and have no observable child feces in the yard (*p* = 0.03) than those with “partial” or “poor” knowledge [5]. In the third district, 80% of CHC members had “full” knowledge of diarrhea prevention and transmission and were significantly more likely to practice 10 WASH behaviors (e.g., covered drinking water, pour to waste for hand washing, use hand washing facility, soap for hand washing, pot rack, garbage pit, use of garbage pit, clean yard, no open defecation, and have a home nutrition garden) than 50% of the comparison sample with “full” knowledge of diarrhea (*p* < 0.0001) [7].

### 3.3. Social Capital

Ten papers described social capital outcomes, a multi-dimensional concept that scholars broadly agree is a by-product of social relationships that can generate positive externalities, including cooperation [32,33,34]. There are two main dimensions of social capital, structural (e.g., social networks and group participation) and cognitive (e.g., trust, reciprocity, cohesion, support). The articles in this review described outcomes from both dimensions. One paper reported these dimensions using both quantitative and qualitative data [9], while the remaining nine described qualitative outcomes [5,6,7,8,13,21,22,27,29]. The majority of the ten studies (*n* = 9) reported themes associated with social bonding or bonding relationships [5,6,7,8,9,13,21,27,29]. Many (*n* = 7) reported changes in social support [5,7,9,13,21,22,29], social cohesion, and/or social solidarity [5,6,7,8,9,22,27]. Thereafter, three papers described the role of peer pressure [9,21,29], two described bridging and linking relationships [9,21], and one study described themes associated with interpersonal trust [9]. Finally, six of the studies reported social capital as an outcome [5,6,7,8,13,22], while four reported social capital as both an outcome and a mediator of change in WASH knowledge [9], WASH behaviors [9,29], and collective action [9,21,27,29].

The one study that quantitatively explored the relationship between a CHC intervention and social capital assessed whether the intervention influenced levels of social capital and whether pre-intervention levels of social capital influenced knowledge and behavioral outcomes within the CHC sample. The author did not find a significant treatment effect of the intervention on CHC member household respondents’ self-reported trust, social support, group participation, or social solidarity factor scores. However, within a sub-sample of CHC member respondents, there was a marginally significant decrease in trust (−0.26, CI (−0.53, 0.02), *p* = 0.07) and social solidarity (−0.27, CI (−0.56, 0.02), *p* = 0.07) factor scores from baseline to final. Further, baseline social solidarity factor scores were found to be associated with a significant increase in average WASH behavioral scores in both the CHC member household respondent (0.23, CI (0.05, 0.41), *p* = 0.01) and the CHC member respondent (0.25, CI (0.03, 0.46), *p* = 0.03) samples. In the same study, program participants qualitatively reported that the CHC intervention increased trust, social bonding, and social solidarity amongst the participants. In turn, members used their social capital to apply social pressure and leverage new relationships with other leaders and communities to facilitate WASH behavioral changes and collective action within their community [9].

The remainder of the studies reported qualitative changes in social capital. For example, a study comparing CHCs to a CLTS intervention in Zimbabwe noted how the CHCs created a dynamic in which social bonds were formed and strengthened, which resulted in increased likelihood that CHC members would work together and provide social support [13]. In Sierra Leone, authors reported that a CHC program helped create communal unity and a collective spirit, which resulted in social healing in post-conflict communities, an increase in decision making by women and the formation of new income generation and social support groups [27]. In Zimbabwe, Haiti, and the Dominican Republic, CHC participants described how they used social pressure to encourage maintenance of newly agreed upon behaviors such as hand washing, environmental management, and latrine use and hygiene [9,21,29]. Finally, two papers described how bridging and linking relationships with other communities and governmental/non-governmental stakeholders provided external support and reinforcement, which facilitated participation in and sustainability of the CHCs and engagement in collective action [9,21].

### 3.4. Collective Action

Ten papers reported outcomes associated with collective action. Two papers provided quantitative evidence [19,22], neither of which reported significant findings. The remaining eight papers provided qualitative data [8,9,16,20,21,23,27,28]. Of these ten papers, eight reported collective action around WASH-specific issues, five on environmental management [8,9,19,22,23], and three on drinking water management [9,16,28], while four reported collective action in other areas [9,20,21,27].

A study of a program in a high-density informal settlement in South Africa reported a 50% reduction in informal dumping sites (from four to two), which were converted into community gardens, after three months of implementation [19]. Similarly, a study in three peri-urban communities in Zimbabwe reported that participants organized 17 clean-up campaigns over one year [22]. Qualitative evidence of CHC participants working together to protect, repair, or improve communal drinking water points was reported by two programs in Haiti [8,9] and one in the Democratic Republic of Congo [28]. Finally, CHC participants were also reported to be more likely to participate in village savings and loans programs in Sierra Leone [27] and Rwanda [20] and to work together to repair roads and street lights in Rwanda [20] and Haiti [9].

### 3.5. Health

Six papers reported health outcomes associated with a CHC program. Four studies reported qualitative evidence [27,28,29,30]. The remaining two studies provided quantitative evidence [12,15], one of which included a comparison sample [12]. In Zimbabwe, a retrospective analysis of data from one clinic that served a district where 80% of households participated in four years of CHC program activities found a 10-fold decrease in all WASH-related communicable diseases (e.g., diarrhea, schistosomiasis, acute respiratory illnesses, and skin diseases) from one year prior to the start of the intervention to four years after the intervention concluded [15]. Conversely, a RCT conducted in Rwanda reported no difference between the three study arms (classic CHC, lite CHC, controls) in health (diarrhea) and anthropomorphic outcomes (height for age, length for age, and weight for height) in children under one, two, and five years of age after a two-year intervention [12].

### 3.6. Cost

Four papers described the cost or cost-effectiveness of CHC interventions, with three studies reporting cost per beneficiary [5,17,18] and one study reporting subjective assessments of program cost-effectiveness [24]. The study with the most robust cost per beneficiary analyses was the original CHC program implemented in Zimbabwe. In this paper, the author reported a cost of USD 0.66 per beneficiary for two years of WASH promotion programming that reached 11,450 participants or 68,700 beneficiaries (11,450 × 6 people per household) in three districts and resulted in the formation of 265 CHCs [5]. One author explored the cost-effectiveness of a CHC program using the full six-month WASH curriculum compared to a CHC program with reduced sessions in one district of Rwanda. This study first compiled cost data and then asked program staff to rate the cost-effectiveness of each arm. The author reported that the program using the full curriculum cost a total of USD 191,017 or USD 3820 per household (50 households), while the reduced session program cost a total of USD 59,815 or USD 1196 per household (50 households). In addition, respondents rated the program using the full curriculum as more cost-effective than the reduced session program (*p* < 0.0001) [24].

## 4. Discussion

In this review, we documented and described the major outcomes of CHC programs focused on WASH promotion. The programs were implemented across various social, cultural, and geographical contexts, primarily in African and Caribbean settings. We found six major outcomes reported in the literature, which align in many ways with the outcomes of other WASH promotion programs and provide insights into the CHC model’s theory of change.

### 4.1. WASH Behaviors

The CHC model has demonstrated consistent impact, sometimes significant, on WASH behaviors. The most consistent behavioral outcome reported in the CHC literature is related to defecation practices (*n* = 15). However, significant results were reported for grouped WASH behaviors (multiple behaviors across five dimensions—sanitation/defecation, hand washing, drinking water, kitchen hygiene, and environmental management), defecation practices (latrine construction, latrine hygiene, and stopping open defecation), drinking water treatment practices, and hand washing behaviors (construction of hand washing facilities and use of soap). Two papers reported a dose response effect, where completion of the full six-month WASH curriculum was associated with increased likelihood of adopting WASH behaviors. Another study found a temporal association where adoption of WASH behaviors consistently increased over time, eventually reaching 100% adoption three years after the intervention ended.

These WASH behavioral outcomes are similar to those of other WASH promotion programs where defecation, hand washing, and water treatment behaviors are the most commonly reported outcomes. Although the results were mixed for all intervention types, a review of WASH promotion models found that community-based interventions achieved the most consistent increases in hand washing and defecation behaviors, while directive hygiene messaging and interventions derived from formal psychosocial theory had mixed, limited, or no evidence of impact on both behaviors [35]. The impact of social marketing on hand washing and defecation practices has been mixed, although these approaches have demonstrated consistent increases in purchase of oral rehydration therapies and point-of-use water treatment technologies [35,36]. Sanitation-specific campaigns have achieved increases in latrine construction and use, with India’s Total Sanitation Campaign, infrastructure interventions, and sanitation education interventions such as CLTS and CHCs achieving the most consistent increases [37]. For CLTS specifically, one review reported a dearth of rigorous quantitative evaluations of behavioral impact but reported significant increases in latrine construction and reductions in open defecation in the available quantitative evaluations [38]. Measurement of kitchen hygiene, environmental management, and grouped WASH behaviors appears unique to evaluations of CHC programs and to our knowledge, no other WASH promotion programs have reported any form of a dose response between participation and WASH behaviors.

### 4.2. WASH Knowledge

Half (*n* = 13) of the studies in this review described increases in WASH knowledge amongst CHC participants, with five reporting significant differences in categorical WASH knowledge scores between CHC participants and comparison samples. Of the papers that provided quantitative results, measures included knowledge of the transmission and prevention of WASH-related diseases (including diarrhea, skin diseases, intestinal worms, and malaria) as well as when to wash hands to prevent diarrhea and how to make a homemade oral rehydration solution. Further, two papers described associations between WASH knowledge and behaviors; CHC participants with higher knowledge scores were more likely to practice preventive WASH behaviors, while a comparison sample with high knowledge scores were less likely to practice WASH behaviors than a CHC sample with high knowledge scores [5,7].

Two prior reviews of WASH promotion programs (including community-based, directive hygiene messaging and psychosocial theory, and social marketing interventions) described the impact of interventions on WASH knowledge. Aligned with our results, one prior review of hand washing and sanitation interventions found that community-based interventions consistently and significantly improved knowledge about key times to wash hands; however, this prior review indicated the impact on knowledge of diarrhea was mixed [35]. This prior review also found no consistent or demonstrable effect of directive hygiene messaging and psychosocial interventions on diarrhea or hand washing knowledge. The impact of social marketing interventions is mixed, with one review reporting no demonstrable effect on diarrhea knowledge and mixed results on hand washing knowledge, while another reported consistent increases in awareness about WASH products, such as oral rehydration solutions and point-of-use water treatment, and hand washing behaviors [35,36].

### 4.3. Social Capital and Collective Action

This review highlighted a potential relationship between social capital and collective action, where CHCs appear to increase social cohesion and social support, which contributes to increased collective action. The type of collective action was mostly associated with communal clean-up campaigns and participation in other community development projects. Qualitative evidence indicates that the CHC model contributes to increased bonding, social support, social solidarity, and women’s empowerment. CLTS programs have reported similar qualitative evidence of increased community mobilization beyond WASH and impacts on women and girls [38]. However, it must be noted that the current evidence regarding social capital and collective action in CHC programs is weak. Of the ten papers that described a relationship with social capital in this review, only one measured social capital quantitatively and the results were inconclusive.

### 4.4. Health

The evidence of health impacts associated with the CHC model is limited. One study using clinical data reported a substantial decrease in WASH-related diseases over a four-year period, while the only RCT in this review found no impact on childhood diarrhea or anthropometrics after two years. This is not surprising as most WASH promotion programs have struggled to demonstrate consistent and significant impacts on health outcomes, especially diarrhea due to limitations of self-report data and temporality. For example, a prior review found that community-based WASH interventions had some, but no consistent, impact on diarrhea, while directive hygiene education and psychosocial interventions had no impact on diarrhea [35]. Social marketing interventions were found to have limited impact on health [35,36]. Finally, CLTS programs reported a similar, inconsistent impact of reductions in open defecation on diarrhea [38].

### 4.5. Program Theory

The evidence we gathered from this review sheds light on the CHC model’s theory of change. The model aims to increase adoption of recommended WASH practices and thus improve health. CHCs achieve this goal by increasing participants’ knowledge and encouraging praxis, or the application of knowledge in daily life [39]. The model theorizes that the gap between WASH knowledge and behavior change is addressed through increased social capital, which facilitates consensus building and changes in communal norms around WASH behaviors and collective action.

This review found both quantitative and qualitative evidence of increased WASH knowledge amongst CHC participants. This is an expected outcome since the WASH curriculum takes six months to complete and all studies measured knowledge immediately after the intervention concluded. Although the evidence is limited, this review also documented significant associations between high WASH knowledge and increased adoption of recommended WASH behaviors amongst CHC participants. Comparison samples with equally high knowledge were less likely to adopt recommended WASH behaviors. This supports the theory that knowledge is a necessary pre-requisite for behavior change but is insufficient by itself.

There is also evidence that CHC programs achieve the goal of increasing adoption of recommended WASH behaviors amongst program participants. The studies in this review reported changes in observable WASH behaviors across five dimensions (e.g., defecation practices, personal hygiene and hand washing, drinking water, environmental management, and kitchen hygiene), which align with the content of the six-month WASH curriculum. Most studies provided quantitative evidence, but significant results were reported in only a handful of studies, most consistently around defecation practices and composite WASH behavior scores. Although determining the factors that led to greater adoption of WASH behaviors was outside the scope of this review, we did find studies that reported evidence of potential dose response and temporal effects of the CHC intervention. These findings lend support to the theory that social processes facilitate behavior change.

Further, the social capital evidence provides some insight into this possible dose response. All of the studies describing social capital provided qualitative evidence that the CHC model increases social bonding, social support, and social cohesion. It is logical that longer engagement with a CHC might lead to greater bonding and cohesion, which should increase trust. Increased trust and cohesion would allow participants to leverage social support and apply social pressure to facilitate behavioral changes. Interestingly, there is evidence that higher levels of pre-intervention social capital are a determinant of greater WASH behavior change, which supports findings from the broader social capital and WASH literature [9,40]. Further, the evidence about collective action might also help explain the observed temporal associations. If CHC participants continue their engagement beyond the six-month WASH curriculum, participants would continue to strengthen their social relationships, thereby facilitating continued and sustained behavioral changes. Much of this is conjecture and the impact of the model on social capital and collective action outcomes and the role of social capital as a moderator of knowledge and behavioral outcomes remains an area for further investigation.

### 4.6. Limitations

This review was limited in that we did not formally rate the quality of the study designs, nor did we consider variation in outcomes based upon country or setting (urban vs. rural). Although desirable, we found such assessments challenging due to the high degree of variability in the study designs and the inconsistent outcome measures reported in the literature. Further, there have been limited peer-reviewed publications describing the impact of CHC programs on WASH outcomes. Much of the evidence is found in the gray literature and in programmatic reports, which are challenging to identify and access. It is possible that we missed some documents in our review.

## 5. Conclusions

This is the first review of the literature describing outcomes associated with global CHC programs focused on WASH promotion. Although there is evidence of an association between CHCs and changes in WASH knowledge and behavior, there is considerable room to strengthen the quality of the research and evaluation of the CHC model globally. Most of the evidence around CHC programs globally is derived from program monitoring data and field reports. There is an urgent need for more rigorous, experimental study designs that can empirically describe the impact of the intervention on both intermediate outcomes (knowledge, behavior) and more distal outcomes (social capital and health). Furthermore, these studies should also include process evaluations in order to capture key factors of implementation that may influence outcomes such as fidelity to the six-month curricula, the amount of resources provided for implementation, methods used to recruit participants, and socio-political contextual factors. Due to the difficulties and costs associated with conducting experimental study designs, we recommend rigorously designed, quasi-experiments using mixed methods research. In addition, we note a need to standardize outcome measures across countries and contexts, especially those related to WASH behaviors and social capital. Such standardization would improve our understanding of program theory and the true impact of the intervention and allow for cross-program assessments.

We also recommend studies that compare the CHC model to other WASH promotion and behavior change interventions. Only one study compared CHCs to CLTS in Zimbabwe. Although the results for the CHC program were encouraging, particularly as they relate to short-term outcomes, this was an isolated study in a country where CHCs have been implemented by the same organization for two decades. Additional side-by-side assessments would help determine the relative strengths and weaknesses of each intervention in achieving knowledge, behavioral, social, and health outcomes. Finally, there is a need for longitudinal studies that measure the sustainability of knowledge and behavioral changes beyond one-year post-implementation and that explore the relationship between sustained behavioral changes and long-term health and social outcomes.

The CHC model is one intervention model in our international toolkit for WASH promotion and behavior change that has demonstrated the potential to achieve community-wide impact. However, despite the relatively widespread adoption of CHC models, ours is the first review to synthesize its potential impact. While the evidence is promising, our understanding of the true impact of the CHC model remains limited by a lack of rigorous research of both processes and outcomes. It is our hope that this review will stimulate interest amongst both WASH practitioners and researchers to further evaluate the potential of this intervention. Furthermore, WASH promotion is only the first step in a multi-stage community development process where the CHC serves as the vehicle for the achievement of wider community development. As we are now in the era of the Sustainable Development Goals, which, in part, challenge the historical focus on vertical (single outcomes) interventions, this model offers the opportunity for practitioners to address multiple health and development determinants.

## Figures and Tables

**Figure 1 ijerph-18-01880-f001:**
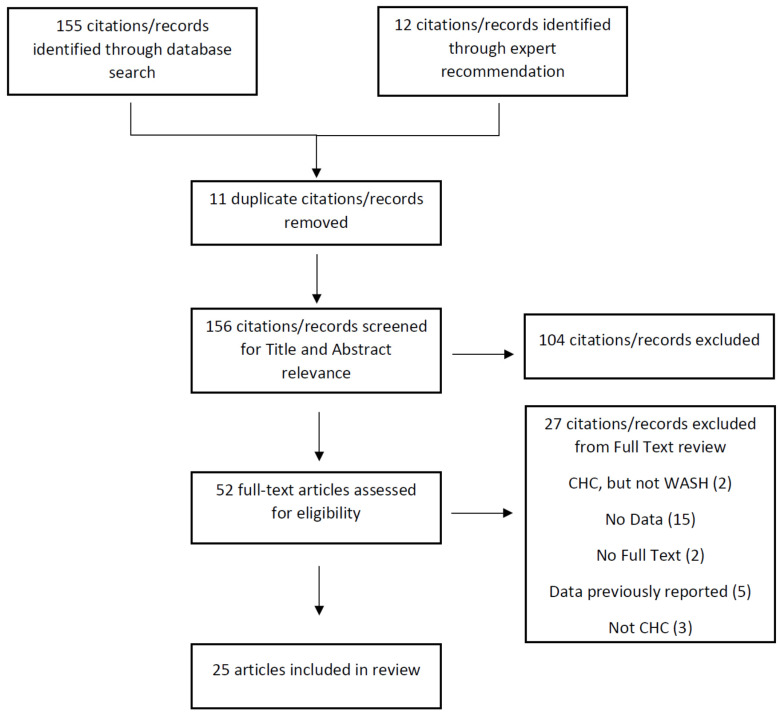
PRISMA Diagram. Note: WASH refers to Water, Sanitation and Hygiene and CHC refers to Community Health Clubs.

**Table 1 ijerph-18-01880-t001:** Community health club (CHC) literature review search terms.

Concept	Key Words/Search Terms
Community Health Club	“Community Health Club” OR “Health Club”
AND	
Water, Sanitation, and Hygiene (WASH)	“water” OR “sanitation” OR “hygiene” OR “WASH” OR “WaSH” OR “WatSan” OR “WATSAN” OR “water and sanitation”

**Table 2 ijerph-18-01880-t002:** Characteristics of reviewed articles.

Author(s)	Type of Article	Study Design	Data Source(s)	Outcome(s)	Countries
Waterkeyn, Okot, and Kwame (2005)	Conference Paper	– Retrospective, cross-sectional (mid-intervention)	– Program monitoring data – Participant observations	WASH Behaviors	Uganda
Waterkeyn and Cairncross (2005)	Peer-Reviewed Manuscript	– Retrospective, cross-sectional (post-intervention) using mixed methods	– Participant surveys – Household observations – Key informant interviews	WASH Behaviors Social	Zimbabwe
Waterkeyn (2005)	Conference Paper	– Retrospective, cross-sectional	– Clinical register data	Health	Zimbabwe
Waterkeyn (2006)	Doctoral Dissertation	– Retrospective, cross-sectional (post-intervention) using mixed methods	– Participant surveys – Household observations – Qualitative interviews – Focus group discussions with pair-wise ranking exercise	WASH Behaviors WASH Knowledge Social Cost	Zimbabwe
Azurduy, Stakem, and Wright (2007)	Program Report	– Retrospective, cross-sectional (post-intervention)	– Participant surveys – Household observations – Qualitative interviews – Focus group discussions	WASH Behaviors WASH Knowledge Social Collective Action Health	Sierra Leone
Rosenfeld (2008)	Conference Paper	– Retrospective, cross-sectional (post-intervention)	– Program monitoring data – Household surveys – Community observations	WASH Behaviors Collective Action	South Africa
Waterkeyn, Matimati, and Muringaniza (2009)	Conference Paper	– Retrospective, cross-sectional (post-intervention)	– Program monitoring data – Household observations	WASH Behaviors Cost	Uganda and Zimbabwe
Waterkeyn and Rosenfeld (2009)	Conference Paper	– Prospective, quasi-experimental (pre- and post-intervention)	– Participant surveys – Household observations	WASH Behaviors Cost	South Africa and Zimbabwe
Maksimoski and Waterkeyn (2010)	Conference Paper	– Prospective, quasi-experimental (pre- and mid-intervention)	– Participant surveys – Household observations – Communal observations	WASH Behaviors Collective Action	South Africa
Whaley and Webster (2011)	Peer-Reviewed Manuscript	– Retrospective, case study of CHC and CLTS interventions	– Participant surveys – Household observations – Qualitative interviews – Focus group discussions	WASH Behaviors Social	Zimbabwe
Ncube (2013)	Master’s Thesis	– Retrospective, cross-sectional (post-intervention) using mixed methods	– Participant surveys – Household observations – Qualitative interviews – Focus group discussions	WASH Behaviors WASH Knowledge Social Collective Action	Zimbabwe
Waterkeyn and Waterkeyn (2013)	Peer-Reviewed Manuscript	– Retrospective, cross-sectional (post-intervention) using mixed methods	– Participant surveys – Household observations – Qualitative interviews – Focus group discussions	WASH Behaviors WASH Knowledge Social	Zimbabwe
Chingono (2013)	Master’s Thesis	– Retrospective, case study (post-intervention) using mixed methods	– Household observations – Qualitative interviews – Focus group discussions	WASH Behaviors WASH Knowledge Social Collective Action	Zimbabwe
Brooks et al. (2015)	Peer-Reviewed Manuscript	– Retrospective, case study (post-intervention) using mixed methods	– Participant surveys – Household observations – Qualitative interviews	WASH Behaviors WASH Knowledge Social Collective Action	Haiti
Rosenfeld and Taylor (2015)	Working Paper	– Prospective, quasi-experimental (pre- and post-intervention) using mixed methods	– Participant surveys – Household observations – Qualitative interviews – Focus group discussions	WASH Behaviors WASH Knowledge	Dominican Republic
Beesley and Feeny (2016a)	Program Report	– Retrospective, case study	– Qualitative interviews	WASH Behaviors WASH Knowledge Collective Action Health	Democratic Republic of Congo
Beesley and Feeny (2016b)	Program Report	– Retrospective, case study	– Qualitative interviews	WASH Behaviors WASH Knowledge Social Health	Democratic Republic of Congo
Beesley et al. (2016)	Program Report	– Retrospective, case study	– Qualitative interviews	WASH Knowledge Health	Democratic Republic of Congo
Ndayambaje (2016)	Master’s Thesis	– Retrospective, cross-sectional (post-intervention)	– Participant surveys – Desk review of government and program data	WASH Behaviors Cost	Rwanda
Munyoro (2016)	Master’s Thesis	– Retrospective, cross-sectional (post-intervention) using qualitative methods	– Household observations – Qualitative interviews – Focus group discussions	WASH Behavior WASH Knowledge Collective Action	Zimbabwe
Sinharoy et al. (2017)	Peer-Reviewed Manuscript	– Prospective, cluster randomized trial (pre- and post-intervention)	– Participant surveys – Household observations – Anthropometrics – Drinking Water Samples	WASH Behaviors Health	Rwanda
Ntakarutimana and Ekane (2017)	Conference Paper	– Retrospective, case–control (post-intervention) using mixed methods	– Participant surveys – Household observations – Qualitative interviews – Focus group discussions – Government data	WASH Behaviors Collective Action	Rwanda
Matimati (2017)	Master’s Thesis	– Retrospective, cross-sectional (post-intervention) using mixed methods	– Participant surveys – Household observations – Focus group discussions	WASH Behaviors WASH Knowledge	Zimbabwe
Pantoglou (2018)	Master’s Thesis	– Retrospective time series analysis	– Program monitoring data – Household observations	WASH Behaviors	Rwanda
Rosenfeld (2019)	Doctoral Dissertation	– Prospective, quasi-experimental (pre- and post-intervention) using mixed methods	– Participant surveys – Household observations – Focus group discussions	WASH Behaviors WASH Knowledge Social Collective Action	Haiti

**Table 3 ijerph-18-01880-t003:** Samples, measures, and results of reviewed articles.

Author(s)	Sample	Reported Measures	Results
**WASH Behaviors**			
Waterkeyn, Okot, and Kwame (2005)	– Census of 15,522 participants from 116 CHCs in 15 Internally Displaced Camps	– Observations of constructed latrines and hygiene facilities	– CHC participants constructed 8583 latrines – CHC participants constructed 6062 bathing shelters – Two camps where all CHC households (100%) constructed pot racks plus spill-over to non-CHC households resulted in a percent increase (above CHC households) in households with pot racks of 159% and 146%
Waterkeyn and Cairncross (2005)	– Survey and observations – Random sample of 736 participants from 50 of 297 CHCs in two districts – Random sample of 172 respondents from 2 matched comparison villages in two districts	– 20 observable indicators of good hygiene practices focused on defecation, drinking water, hand washing, kitchen hygiene, and environmental hygiene behaviors	– District 1: significant differences between intervention and comparison households on 16 WASH behaviors (*p* < 0.001) – District 2: significant differences between intervention and comparison households on 9 WASH behaviors (*p* < 0.01)
Waterkeyn (2006)	– Random sample of 736 participants from 50 of 297 CHCs in two districts – Random sample of 172 respondents from 2 matched comparison villages in two districts	– 20 observable indicators of good hygiene practices focused on defecation, drinking water, hand washing, kitchen hygiene, and environmental hygiene behaviors	– Significant differences between intervention and comparison households on 20 WASH behaviors (*p* < 0.01)
Azurduy, Stakem, and Wright (2007)	– Purposive sample of participants from 7 of 56 CHCs, program staff, and community leadership in 1 district – Purposive sample of respondents from 5 comparison communities in 1 district	– Self-reported childcare practices and observations of household environment	– CHC participants observed to have more clotheslines, pot racks, and cleaner home environments than comparison sample
Rosenfeld (2008)	– Census of 995 participants from 9 CHCs in 1 rural municipality	– Self-reported drinking water treatment behaviors	– 59% of participants reported treating their drinking water at home by boiling or using chlorine after 6 months (41% increase from baseline)
Waterkeyn, Matimati, and Muringaniza (2009)	– Census of 14,282 participants from 116 CHCs in 15 Internally Displaced Camps	– Observations of constructed latrines and hygiene facilities	– CHC participants constructed 11,932 latrines – 58% of CHC participants constructed pot racks – 43% of CHC participants constructed bathing shelters – 25% of CHC participants constructed hand washing facilities
Waterkeyn and Rosenfeld (2009)	– 2501 participants from 37 CHCs in Zimbabwe – 311 participants from 3 CHCs in South Africa	– 17 observable indicators (Zimbabwe) and 12 observable indicators (South Africa) of good hygiene practices focused on defecation, drinking water, hand washing, kitchen hygiene, and environmental hygiene behaviors	– 80% (44% average change from baseline to final) of CHC participants practiced 17 observable WASH behaviors in Zimbabwe – 76% (36% average change from baseline to final) of CHC participants practiced 12 observable WASH behaviors in South Africa
Maksimoski and Waterkeyn (2010)	– Random sample of 89 heads of household in 1 community – 52 participants (census) from 1 CHC in 1 community	– 10 observable indicators of household health, sanitation, and hygiene	– 75.6% increase in CHC households reporting zero open defecation at mid-line – 79.2% of CHC households categorized as practicing high WASH behaviors compared to 36.9% of non-CHC households at mid-line
Whaley and Webster (2011)	– Random sample of 115 participants from 2 randomly sampled CHC communities in 2 districts – Random sample of 118 participants from 1 randomly sampled and 1 purposively sampled CLTS communities in 2 districts	– 11 observable indicators of good hygiene practices focused on defecation and hand washing	– Households in CHC communities had significantly greater reduction in open defecation and use of hand washing facilities compared to CLTS communities (*p* < 0.0001) – Households in CLTS communities more likely to have a latrine than CHC communities (44% vs. 26%) – CHC households more likely to sustain use of hand washing facilities than CLTS households (37% vs. 2%)
Ncube (2013)	– Random sample of 175 participants from 3 CHCs in 1 peri-urban district – Random sample of 60 respondents from 1 comparison community in 1 peri-urban district	– Self-reported defecation and hand washing practices – Observations of household hygiene, latrines, drinking water, and hand washing behaviors	– 30–40% increase in observable clean yards, toilets, and water points – 92% of CHC respondents correctly demonstrated hand washing using pour to waste compared to 35% of comparison respondents
Waterkeyn and Waterkeyn (2013)	– Random sample of 1124 participants from 76 of 382 CHCs in 3 districts – Random sample of 276 respondents from 3 matched comparison villages in 3 districts	– 10 observable indicators of good hygiene practices focused on defecation, drinking water, hand washing, kitchen hygiene, and environmental hygiene behaviors	– CHC participants were significantly more likely to practice 10 WASH behaviors than the comparison group (*p* < 0.001) – 93.4% of CHC participants practiced safe sanitation compared to 43.2% of comparison sample (*p* < 0.001)
Chingono (2013)	– Random sample of 60 participants from 6 of 39 CHCs in 1 district – Random sample of 20 respondents from 2 comparison villages in 1 district	– Observations of household drinking water, defecation, and hand washing behaviors	– CHC participants demonstrated a 15% increase in use of borehole water, 18% increase in latrine ownership, and 22% increase in the presence of hand washing facilities – The majority of CHC households observed to have clotheslines and pot racks
Brooks et al. (2015)	– Census of 52 participants from 3 of 23 purposively sampled urban CHCs – Random sample of 146 non-CHC heads of household from 3 purposively sampled urban neighborhoods	– Self-reported and household observations of drinking water, defecation, hand washing, and environmental management practices	– Comparison respondents were 7.1 times more likely to report open defecation than CHC respondents (*p* < 0.02) – CHC participants were more likely to practice improved hand washing, drinking water storage, and environmental management practices than comparison respondents
Rosenfeld and Taylor (2015)	– Random sample of households (participants and non-participants) in 5 communities with CHC intervention	– 20 observable indicators of good hygiene practices focused on defecation, drinking water, hand washing, kitchen hygiene, and environmental hygiene behaviors	– CHC participants showed no significant change in observable behaviors from baseline to final – No significant difference in observable behaviors between CHC participants and comparison respondents at final
Beesley and Feeny (2016a)	– Participants from 1 CHC in 1 rural village – Program staff	– Perceived changes in participants’ water, sanitation, and hygiene behaviors	– CHC participants reported constructing garbage pits and latrines with hand washing facilities
Beesley and Feeny (2016b)	– Participants from 1 CHC in 1 rural village – Program staff	– Perceived changes in participants’ water, sanitation, and hygiene behaviors	– CHC participants reported construction of latrines, hand washing facilities, garbage pits, and pot racks – CHC participants reported improved kitchen hygiene practices
Ndayambaje (2016)	– Random sample of 50 participants from 1 “classic” CHC in 1 village from 1 district – Random sample of 50 participants from 1 “lite” CHC in 1 village from 1 district – Purposive sample of 44 program administrators and trainers	– 10 observable indicators of household water, sanitation, and hygiene practices – Perceived impact of “classic” and “lite” arms on household hygiene, waste management, environmental management, community wellness, malaria control, and drinking water practices – Perceived effectiveness of “classic” and “lite” arms on household hygiene, waste management, environmental management, community wellness, malaria control, and drinking water practices	– Greater improvement in WASH behaviors from baseline to final in the “classic” arm than the “lite” arm: hygienic latrine (14.4% vs. 2.4%), hand washing facility (41% vs. 5.1%), household water treatment (15.6% vs. 3.7%) – CHC participants (classic and lite) rated the intervention as having the greatest impact on malaria control (use of mosquito nets and treatment), community wellness (participate in wellness programs), and household hygiene (hand washing facilities) practices – Key informants rated the “classic” arm as more effective than the “lite” arm on achieving change in all behavioral outcomes
Munyoro (2016)	– Purposive sample of 15 participants from 6 of 12 CHCs in 1 urban area – Convenience sample of 90 project staff and town leaders	– Perceived changes in participant WASH behaviors	– Respondents reported increased personal hygiene practices, including brushing teeth, combing hair, bathing, washing clothes, and cutting nails – Respondents observed to stop using the common bowl method of hand washing and begin using the pour to waste method – CHC participants observed to increase storing drinking water in covered containers – CHC households observed to have garbage pits, compost pits, clean latrines, and clean yards after the intervention
Sinharoy et al. (2017)	– Random sample of 2729 participants from 50 “classic” CHCs with children under 5 years in 1 district – Random sample of 2482 participants from 50 “lite” CHCs with children under 5 years in 1 district – Random sample of 2723 respondents from 50 control communities with children under 5 years in 1 district	– Observations of household latrines and hand washing facilities – Self-reported drinking water source, drinking water treatment, child feces disposal practices – Colony-forming units of fecal coliforms per 100 mL of water	– Households in the “classic” CHC arm were significantly more likely to treat their drinking water (*p* = 0.003) and have a latrine than control households (*p* = 0.017) – Participants in the “classic” arm who completed 20 CHC sessions were significantly more likely to report treating their drinking water and have a structurally complete latrine than controls – No significant differences in behaviors between households in the “lite” CHC arm and control households
Ntakarutimana and Ekane (2017)	– Random sample of 407 participants from 2 CHCs (1 peri-urban and 1 rural) in 2 districts – Random sample of 391 respondents from 2 control communities (1 peri-urban and 1 rural) in 2 districts	– Observations of household latrines and hand washing practices	– Peri-urban (*p* = 0.0001) CHC households were significantly more likely at final to have an improved toilet (89.4% vs. 74.2%), clean toilet (69.5% vs. 28%), functional hand washing facility (74.2% vs. 13.7%), and soap (38.4% vs. 7.7%) than controls – Rural (*p* = 0.0001) CHC households were significantly more likely at final to have an improved toilet (95.2% vs. 14.2%), clean toilet (98.1% vs. 45%), functional hand washing facility (91.4% vs. 43.3%), and soap (92.4% vs. 4.2%) than controls
Matimati (2017)	– Random sample of 30 adult household members from 10 communities with a CHC in 1 district	– Self-report and observations of household drinking water, kitchen hygiene, defecation, hand washing, and solid waste management practices	– Statistically significant associations between the number of CHC sessions attended and treating drinking water (*p* < 0.0001), having a clean toilet (*p* = 0.001), and using soap to wash hands (*p* < 0.0001) – 50% of CHC participants reported behavioral changes were sustained 2 years after the intervention completed
Pantoglou (2018)	– Census of CHC 381 participants from 50 communities receiving the “classic” intervention	– Average scores on 29 observable WASH indicators grouped into 8 main indicators of household hygiene, drinking water source, drinking water storage, hand washing, sanitation, body hygiene, cooking, and childcare at five time points	– Statistically significant improvement in observable hygiene indicators from baseline to: mid-line (*p* = 0.01), end-line (*p* < 0.05), post-intervention I (*p* < 0.05), and post-intervention II (*p* < 0.05) – At post-intervention I and II, 86% and 100% of all recommended practices were observed in sampled households
Rosenfeld (2019)	– Random sample of 381 (baseline) and 284 (final) adult heads of CHC participant households from 15 of 35 randomly sampled CHC communities across 4 communes – Random sample of 326 (baseline) and 237 (final) adult heads of household from 6 matched comparison communities across 4 communes – Purposive sample of 32 CHC participants and 4 CHC facilitators from 4 purposively sampled CHCs (2 high and 2 low change in knowledge and behavior scores) – Purposive sample of 7 program managers and coordinators	– Hygiene index scores (0–14 points) comprised of 16 observable indicators of household WASH practices in five domains: drinking water, sanitation and defecation, hand washing, kitchen hygiene, and environmental/solid waste management practices – Qualitative themes about WASH behaviors and factors that facilitated behavior change	– No significant treatment effect on WASH behavioral scores (*p* = 0.80) – Discussants described how defecation and hand washing behavioral changes were influenced by the knowledge they gained about the link between disease (diarrhea and cholera) and WASH behaviors – Discussants described how behavioral changes became habitual when people realized they avoided diseases such as cholera
**WASH Knowledge**			
Waterkeyn (2006)	– Random sample of 736 participants from 50 of 297 CHCs in two districts – Random sample of 172 respondents from 2 matched comparison villages in two districts	– Quantitative measure of participant knowledge of recipe for homemade oral rehydration solution, proper childcare, and prevention of diarrhea, malaria, bilharzia, worms, skin diseases, HIV/AIDS, and TB	– CHC participants provided significantly higher number of correct responses on 9 questions than comparison respondents (*p* < 0.0001)
Azurduy, Stakem, and Wright (2007)	– Purposive sample of participants from 7 of 56 CHCs, program staff, and community leadership in 1 district – Purposive sample of respondents from 5 comparison communities in 1 district	– Descriptions of knowledge gained through participation	– CHC participants were able to list all topics learned from the curriculum
Ncube (2013)	– Random sample of 175 participants from 3 CHCs in 1 peri-urban district – Random sample of 60 respondents from 1 comparison community in 1 peri-urban district	– Quantitative measures of participant knowledge of oral rehydration solution recipe, childcare, diarrhea, malaria, bilharzia, worms, skin diseases, and HIV/AIDS	– 65% of CHC participants had “good” WASH knowledge, while 65% of comparison respondents had “poor” WASH knowledge
Waterkeyn and Waterkeyn (2013)	– Random sample of 1124 participants from 76 of 382 CHCs in 3 districts – Random sample of 276 respondents from 3 matched comparison villages in 3 districts	– Quantitative measures of participant knowledge about appropriate childcare and the transmission and prevention of diarrhea, schistosomiasis, worms, skin diseases, malaria, HIV/AIDS, and TB – 10 observable indicators of good hygiene practices	– 68.3% of CHC participants demonstrated “full knowledge” of diarrhea compared to 38.2% of comparison respondents (*p* < 0.001) – 80% of CHC participants practicing 10 recommended WASH behaviors demonstrated “full knowledge” of diarrhea compared to 50% of comparison respondents – A greater proportion of CHC participants demonstrated full knowledge of all topics than comparison respondents (20% average difference on all topics)
Chingono (2013)	– Random sample of 60 participants from 6 of 39 CHCs in 1 district – Random sample of 20 respondents from 2 comparison villages in 1 district	– Self-reported descriptions of knowledge about the CHC curriculum	– CHC participants reported increased knowledge about disease management, nutrition, personal hygiene, environmental hygiene, and child health (e.g., vaccinations, growth monitoring, and exclusive breastfeeding)
Brooks et al. (2015)	– Census of 52 participants from 3 of 23 purposively sampled urban CHCs – Random sample of 146 non-CHC heads of household from 3 purposively sampled urban neighborhoods	– Aggregated scores (number of correct responses categorized as low, medium low, medium high, and high) measuring participant knowledge of hand washing, diarrhea, skin diseases, worms, malaria, and dengue	– CHC participants were significantly more likely to have high preventive WASH knowledge scores (71.2% vs. 4.1%) compared to comparison respondents (*p* < 0.0001)
Rosenfeld and Taylor (2015)	– Random sample of households (participants and non-participants) in 5 communities with CHC intervention	– Total correct responses to questions measuring knowledge of hand washing, diarrhea, skin diseases, worms, and dengue	– CHC participants knowledge scores increased significantly from baseline to final – CHC participants knowledge scores at final were significantly higher than comparison respondent scores
Beesley and Feeny (2016a)	– Participants from 1 CHC in 1 rural village – Program staff	– Perceived changes in participants’ water, sanitation, and hygiene knowledge	– CHC participants reported increased awareness about the importance of hygiene practices to prevent disease
Beesley and Feeny (2016b)	– Participants from 1 CHC in 1 rural village – Program staff	– Perceived changes in participants’ water, sanitation, and hygiene knowledge	– CHC participants reported increased knowledge about nutrition and kitchen hygiene
Beesley et al. (2016)	– Participants from 1 CHC in 1 rural village – Program staff	– Perceived changes in participants’ water, sanitation, and hygiene knowledge	– CHC participants reported increased knowledge about nutrition, kitchen hygiene, and personal hygiene
Munyoro (2016)	– Purposive sample of 15 participants from 6 of 12 CHCs in 1 urban area – Convenience sample of 90 project staff and town leaders	– Perceived changes in participant knowledge	– Respondents described increased knowledge about WASH diseases and the importance of personal hygiene, hand washing, and safe drinking water – Respondents described increased knowledge about diarrhea, malaria, bilharzia, worms, TB, dysentery, and HIV/AIDS – A significant increase in the number of participants who could name the causes and prevention of diarrhea was reported
Matimati (2017)	– Purposive sample of 43 participants and leaders from 10 CHC communities	– Perceived changes in the community	– CHC participants described increases in knowledge about WASH diseases and disease prevention
Rosenfeld (2019)	– Random sample of 381 (baseline) and 284 (final) adult heads of CHC participant households from 15 of 35 randomly sampled CHC communities across 4 communes – Random sample of 326 (baseline) and 237 (final) adult heads of household from 6 matched comparison communities across 4 communes – Purposive sample of 32 CHC participants and 4 CHC facilitators from 4 purposively sampled CHCs (2 high and 2 low change in knowledge and behavior scores) – Purposive sample of 7 program managers and coordinators	– Composite knowledge score (0–26 points) comprised of the total number of correct responses to four questions about diarrhea transmission, when to wash hands, prevention of skin diseases, and the ingredients for homemade oral rehydration solution – Qualitative themes focused on participant learning and information dissemination through the CHC	– Significant treatment effect on composite WASH knowledge scores (*p* < 0.0001) – Discussants described how the focus of the intervention was to increase participants’ knowledge and disseminate information through the community – Discussants described how knowledge about diseases such as cholera led to WASH behavioral changes
**Social Capital**			
Waterkeyn and Cairncross (2005)	– Purposive sample of 20 participants from 10 CHCs in 1 district	– Perceived personal and social impact of the intervention	– CHC participants describe increased self-confidence, social bonding, social standing, and respect from husbands
Waterkeyn (2006)	– Census of participants from 10 CHCs in 1 district – Purposive sample of 70 participants from 10 CHCs in 1 district	– Self-reported reasons for participation and perceived personal and social impact of the intervention	– CHC participants described increased social bonding, social standing, and respect as a result of participating in the intervention – Pair-wise ranking exercises revealed that the third most valued impact of the CHC was the creation of a sense of belonging
Azurduy, Stakem, and Wright (2007)	– Purposive sample of participants from 7 of 56 CHCs, program staff, and community leadership in 1 district – Purposive sample of respondents from 5 comparison communities in 1 district	– Perceived social impact of the intervention	– CHC participants described increased collective spirit, unity, and women making decisions as a result of participating in the intervention
Whaley and Webster (2011)	– Purposive sample of 13 participants from 3 CHCs in 2 districts – Purposive sample of 12 CLTS participants from 4 communities in 1 district – Purposive sample of 12 CHC and CLTS program staff	– Factors influencing participation and behavior change	– CHCs contributed to the formation and strengthening of social bonds where participants reported they were more likely to help each other
Ncube (2013)	– Purposive sample of 60 participants from 3 CHCs in 1 peri-urban district – Purposive sample of 15 district leaders in 1 peri-urban district	– Factors influencing sustainability, participation, and relevance of the intervention	– Key informants described increased social cohesion amongst CHC participants, including more social support for members facing difficult situations such as a death in the family
Waterkeyn and Waterkeyn (2013)	– Census of 750 participants from 10 CHCs in 1 district – Purposive sample of 20 participants from 10 CHCS in 1 district	– Perceived changes in participants’ lives since joining the CHC – Reasons participants enjoyed the CHC	– CHC participants in group discussions ranked themes related to a “Need for Belonging” (social inclusion, social support, consensus) as the third most important change in their life – CHC interviewees reported themes related to “social interaction” as the second most common reason they liked the CHCs
Chingono (2013)	– Purposive sample of CHC participants, program staff, health outreach workers, and local leadership	– Perceived impact on social cohesion, social support, women’s roles, and engagement with health and development agencies	– Key informants and discussants reported increased social bonding, social support, and women’s participation in decision making and leadership – Key informants and discussants reported the importance of linking relationships and social pressure – Key informants and discussants reported increased coordination between CHC communities and the formal health sector
Brooks et al. (2015)	– Census of 17 CHC facilitators and supervisors in Port-au-Prince	– Perceived role of collective identity formation on social cohesion	– Key informants reported an increase in social bonding and cohesion amongst CHC participants as a result of collective identity formation
Beesley and Feeny (2016b)	– Participants from 1 CHC in 1 rural village – Program staff	– Perceived impact on social aspects of community life	– CHC participants report increased social support, bonding, and social pressure to adhere to behavioral changes
Rosenfeld (2019)	– Random sample of 381 (baseline) and 284 (final) adult heads of CHC participant households from 15 of 35 randomly sampled CHC communities across 4 communes – Random sample of 326 (baseline) and 237 (final) adult heads of household from 6 matched comparison communities across 4 communes – Purposive sample of 32 CHC participants and 4 CHC facilitators from 4 purposively sampled CHCs (2 high and 2 low change in knowledge and behavior scores) – Purposive sample of 7 program managers and coordinators	– 19 social capital items from the World Bank Social Capital Assessment Tool reduced to four principle factor scores: group participation, social support, trust, and social solidarity – Qualitative themes describing the impact of the CHC intervention on social capital factors and the role social capital factors played in facilitating or influencing knowledge dissemination, behavior change, and collective action	– No significant treatment effect on social capital factor scores (trust, social support, participation, social solidarity) – Baseline social solidarity factor scores associated with a significant increase in average WASH behavior scores from baseline to final (*p* = 0.01) – There was a marginally significant interaction between the intervention and participation scores on average WASH knowledge scores (*p* = 0.08), and a significant interaction between the intervention and social solidarity scores on average hygiene index scores (*p* = 0.04). – Discussants reported the intervention increased trust, social bonding, and social solidarity – Discussants described how social pressure, social solidarity, and bridging relationships with other clubs facilitated WASH behavioral changes and engagement in collective action – Communities with low trust, weak social solidarity, and limited social networks achieved lower degrees of WASH behavior change
**Collective Action**			
Azurduy, Stakem, and Wright (2007)	– Purposive sample of participants from 7 of 56 CHCs, program staff, and community leadership in 1 district – Purposive sample of respondents from 5 comparison communities in 1 district	– Perceived changes in collective action	– CHC participants reported they are more likely to work together after the intervention, specifically to improve roads, conduct outreach education to neighboring communities, and initiate village savings and loan clubs
Rosenfeld (2008)	– Communal drinking water points for 3 of 9 rural CHCs in 1 municipality	– Observations of communal water points before and after the intervention	– CHC participants worked together to improve and protect communal water points using resources available in the community
Maksimoski and Waterkeyn (2010)	– Observations by program staff and evaluator	– Communal observations and participant self-reports	– 50% reduction in informal dumping sites, with two converted into communal gardens – CHC participants worked together to clean communal latrines and ablution blocks
Ncube (2013)	– Random sample of 175 participants from 3 CHCs in 1 peri-urban district – Random sample of 60 respondents from 1 comparison community in 1 peri-urban district	– Self-reported and observed communal clean-up campaigns	– CHC participants engaged in 17 community cleanliness campaigns during the intervention period
Chingono (2013)	– Purposive sample of CHC participants, program staff, health outreach workers, and local leadership	– Perceived impact on engagement in collective activities	– Key informants and discussants reported CHC participants are more likely to self-initiate projects to improve collective well-being, including establishing communal gardens and joining village savings and loan programs
Brooks et al. (2015)	– Census of 17 CHC facilitators and supervisors in Port-au-Prince	– Descriptions of CHC participants working together	– Key informants describe CHC participants working together to clean neighborhoods and remove standing water
Beesley and Feeny (2016a)	– Participants from 1 CHC in 1 rural village – Program staff	– Perceived impact on engagement in collective activities	– CHC participants reported working together with NGO partner to construct new safe drinking water points
Munyoro (2016)	– Purposive sample of 15 participants from 6 of 12 CHCs in 1 urban area – Convenience sample of 90 project staff and town leaders	– Reported changes in collective action around solid waste management and illegal dumping	– CHC participants reported engaging in community-wide garbage clean-up campaigns and rehabilitating open spaces that had been converted into garbage dumping sites
Ntakarutimana and Ekane (2017)	– Purposive sample of community leaders, opinion leaders, and community members	– Perceived impact on engagement in collective activities	– Key informants reported CHCs worked together to improve roads and participate in village savings and loan programs
Rosenfeld (2019)	– Purposive sample of 32 CHC participants and 4 CHC facilitators from 4 purposively sampled CHCs (2 high and 2 low change in knowledge and behavior scores) – Purposive sample of 7 program managers and coordinators	– Qualitative themes describing the impact of the CHC intervention on collective action – Qualitative themes describing the role social capital factors played in facilitating or influencing collective action	– Discussants reported the intervention increased collective action in community development activities such as community clean-up campaigns, water point repairs, road repairs, and provision of street lights – Discussants described how increases in collective action were facilitated by enhanced trust, social solidarity, and positive peer pressure
**Health**			
Waterkeyn (2005)	– All patients in clinical registers from two rural clinics serving CHC intervention areas between 1995 and 2004	– Annual WASH-related diseases including diarrhea, skin diseases, and acute respiratory illnesses	– 10-fold decrease in all WASH-related communicable diseases in one clinic where 80% of households in the ward participated from pre-intervention to 4 years post-intervention
Azurduy, Stakem, and Wright (2007)	– Purposive sample of participants from 7 of 56 CHCs, program staff, and community leadership in 1 district – Purposive sample of respondents from 5 comparison communities in 1 district	– Perceived changes in health care utilization and mortality	– CHC participants increased engagement with formal health care and reported reductions in maternal and child mortality
Beesley and Feeny (2016a)	– Participants from 1 CHC in 1 rural village – Program staff	– Perceived impact on health of participants and their family	– CHC participants reported improved health and well-being, with a reduction in disease
Beesley and Feeny (2016b)	– Participants from 1 CHC in 1 rural village – Program staff	– Perceived impact on health of participants and their family	– CHC participants reported a reduction in diseases and deaths from sanitation and hygiene
Beesley et al. (2016)	– Participants from 1 CHC in 1 rural village – Program staff	– Perceived impact on health of participants and their family	– CHC participants reported a reduction in diseases and deaths from sanitation and hygiene
Sinharoy et al. (2017)	– Random sample of 2729 participants from 50 “classic” CHCs with children under 5 years in 1 district – Random sample of 2482 participants from 50 “lite” CHCs with children under 5 years in 1 district – Random sample of 2723 respondents from 50 control communities with children under 5 years in 1 district	– Caregiver-reported diarrhea within the previous 7 days in children under 5 years – Weight for age Z scores, height for age Z scores, and stunting and wasting for children under 5 years	– No measurable differences in diarrhea and anthropometry in children under 5 years between study arms
**Cost**			
Waterkeyn (2006)	– Program data	– Cost per beneficiary	– Estimated cost of USD 0.35 per beneficiary
Waterkeyn, Matimati, and Muringaniza (2009)	– Program data	– Cost per beneficiary	– Estimated cost of USD 0.76 per beneficiary
Waterkeyn and Rosenfeld (2009)	– Program data from Zimbabwe and South Africa CHCs	– Cost per beneficiary	– Estimated cost of USD 3.30 per beneficiary in Zimbabwe – Estimated cost of USD 28.00 per beneficiary in South Africa
Ndayambaje (2016)	– Purposive sample of program administrators and trainers	– Actual costs of the “classic” and “lite” intervention arms and perceived cost-effectiveness of each arm	– The “classic” arm cost USD 3820 per household compared to USD 1196 per household in the “lite” arm – Respondents rated the “classic” arm as more “cost-effective” than the “lite” arm

## Data Availability

No new data were created or analyzed in this study. Data sharing is not applicable to this article.

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
