# Peer review of "A Review of the Community Health Club Literature Describing Water, Sanitation, and Hygiene Outcomes"

_ijerph, 2021, doi:10.3390/ijerph18041880_

Round 1
Reviewer 1 Report
Please see the attached pdf.

Author Response
Thank you for reviewing our manuscript. Here are our responses to your comments/suggestions.
- More concise title - We have rewritten our title in what we feel is a more concise manner: A review of the Community Health Club literature describing water, sanitation and hygiene outcomes
- In the abstract, you asked for clarity about the inclusion criteria for study exclusion and inclusion. We felt that this level of detail was not appropriate/necessary in the abstract and was not feasible considering the word count limitations.
- In the abstract, you asked why we combined knowledge and behaviors. We did not combine these outcomes into one and we hope it is clear from our results section that we treated both outcomes separately. We wrote it this way to be more concise (avoiding writing WASH twice in a row).
- You made note to our in-text citations. We have thoroughly reviewed all of our intext citations and ensured they are in alignment with the journal's required citation style.
- In the background you asked for more detail on why the CHC model holds promise, as we claim at the conclusion of the first paragraph. We feel that this is sufficiently explained in subsequent paragraphs of the introduction, as well as through the results of the review.
- In the background you asked why the program has only been implemented and evaluated at scale in two countries? Our response is that this is due to the limited number of peer reviewed articles describing the CHC model, including any systematic reviews (which this manuscript addresses) AND other systemic/funding issues. However, addressing this is beyond the scope of this manuscript.
- In the background you requested a stronger link between the second to last and last paragraphs. We did not feel this link required further strengthening. The second to last paragraph describes the extent to which the model has been implemented over a 20 year period, while the last paragraph discusses how despite this relatively widespread utilization, no systematic review of WASH outcomes has been performed.
- In the background you requested a stronger expression of the significance of the research. We agreed and wrote two additional sentences to conclude our introduction describing this significance for practicioners and researchers alike.
Reviewer 2 Report
The paper represents a laudable effort to provide an insight into the usefulness of the CHC approach to hygiene and health improvement. It clearly shows that CHC is no ”silver bullet” in that case, which comes as no surprise. Development outcomes have consistently shown themselves to depend heavily on the context, which together with process is the most important determinant. I found some 2005 guidelines related to an application in Sierra Leone, which i.a. state that:
”The most important aspect is to maintain an empowering, inclusive, fun and participatory approach. This Manual is about instilling the CHWs with the essential skills, principles, values, attitudes and behaviours necessary to achieve real community empowerment through the health club approach. This can only be done through experiential, hands-on learning with communities.”
I understand that it has been difficult to account for context and process. In the paper, you often mention a 6 month curriculum. To me, that implies a rigid programme structure, that might affect the effectiveness of a health club.
It seems reasonable to assume that outcomes vary between countries, and also depends on the size of the community, where the intervention takes place. The methods used to recruit participants could have implications.
Have outcomes improved with time? Or, have they rather declined as CHC has become routine, and maybe also proliferated? After all, CHC has a 25 year history by now.
I mention these things, as I think that your presentation of the results has to much of a list structure. With a more organized structure, I think that you may be able to pinpoint some notable facts.
Your paper is interesting and useful. I still think that you might be able able to make it even better through some additional considerations.
Author Response
Thank you for reviewing our manuscript. After reading your comments, we identified the following two principle recommendations for revisions. Please see our responses.
- You asked whether the outcomes associated with the intervention have improved with time? We are also intrigued by this question, but this is beyond the scope of this review. Further, as our manuscript highlights, few studies utilized a longitudinal design that could measure outcomes beyond the immediate conclusion of the intervention.
- You also appeared to suggest changes to our results section. However, without more specific guidance or alternative suggestions, we believe the structure of our results section is appropriate and aligns with the structure of similar systematic reviews. We have decided to leave our results section as is.
Reviewer 3 Report
The community health clubs have worked with WASH activity for some years and different projects have financed these activities, anyhow, the money given to these activities are been rather low in most cases and usually also the local people have given at least their work and some money for these activities.
This review work is interesting. This work has been mainly done in Africa, which is ok. I would had added one sentence about this. The other possibility is to tell that the article is mainly basing on works done in African and Caribbean, since there are areas in other continents (in Europe, Asia, Oceania, Americas) where the similar work should be done, too. The introduction is good and it shows that there are still death cases of small children which could mainly be avoided with very low amount of costs and better knowledge of mothers and other related people.
The content is important. Technically it would be better to divide the Table 3 to six smaller tables as Table 3 a Samples, measured and results of behaviors according to articles of Table 2.
In some cases it looks like the use of soap in hand washing had a higher influence in rural areas than in suburban areas compared not to use soap. In suburban areas families may buy the soap and the price is not cheap so there might be times, that there is no soap. In rural areas people could make soap from ash lime and animal or vegetable fats. People may be able to use saponin containing plants. If the work is not considered, the rural soap was free of charge and thus soap could be really available. In areas where washing is the work belonging to women, they possible had to make also soap and they may have been willing to do this.
The social capital is interesting. In many areas different people may belong to different ethnic groups or they may speak different language or dialect or they may have different religions or political opinions or economic status and education. Could these explain that the collective actions are not easy?
The different cost may depend also on what is included to the needs.
See still once the instructions given to authors and correct all references according to these.
Author Response
Thank you for reviewing our manuscript. We have responded to the following comments and suggestions for revisions.
- You suggested dividing Table 3 into six smaller tables describing the literature associated with each outcome separately. We appreciate this suggestion, but decided that it was more efficient to keep this information in one table.
- You posed the question in the discussion section as to whether variations in ethnic groups and culture could explain the lack of collective action. We are also intrigued by the reasons behind the data we presented and feel that your insights are likely correct. However, including such commentary and interpretation is beyond the scope of this review. We did not attempt to explain why certain outcomes were achieved or not achieved, but rather to describe the common outcomes and compare those outcomes to other WASH promotion interventions.
- Your last suggestion was to check our references and intext citations. We have thoroughly reviewed our citations and ensured that they align with the journals required format.